# Dietary Habits and Nutritional Status in Ecuatorian Children Aged 1–11 Years: A Systematic Review Highlighting the Dual Burden of Malnutrition

**DOI:** 10.3390/nu17223608

**Published:** 2025-11-19

**Authors:** Keila S. Micoanski, Cristina Izquierdo-García, Alex S. Huacho-Jácome, María Trelis, Mónica Gozalbo

**Affiliations:** 1Area of Nutrition and Bromatology, Department of Medicine and Public Health, Food Science, Toxicology, and Legal Medicine, Faculty of Pharmacy and Food Sciences, University of Valencia, Av. Vicent Andrés Estellés, 22, 46010 Burjassot, Spain; 2Area of Parasitology, Department of Pharmacy and Pharmaceutical Technology and Parasitology, Faculty of Pharmacy and Food Sciences, University of Valencia, Av. Vicent Andrés Estellés, 22, 46010 Burjassot, Spainmaria.trelis@uv.es (M.T.); 3Faculty of Public Health, Escuela Superior Politécnica de Chimborazo, Riobamba 060101, Ecuador

**Keywords:** eating habits, Ecuador, childhood, nutritional status, systematic review

## Abstract

**Background:** Background: Childhood dietary habits are critical determinants of physical growth, cognitive development, and long-term health. In Ecuador, malnutrition remains a major public health concern, with both undernutrition and overweight/obesity affecting children—especially in rural and indigenous populations. **Methods:** This systematic review followed PRISMA 2020 guidelines (PROSPERO ID: CRD420251080987). Searches were conducted in SciELO, Dialnet, and ScienceDirect (accessed August 2025) using Boolean operators to identify quantitative studies in Spanish or English published between 2018 and 2025 that assessed dietary habits and nutritional status in Ecuadorian children aged 1–11 years. Seventeen studies, including approximately 12,000 children, were included. **Results:** Prevalence of chronic undernutrition ranged from 15% to 35%, while overweight and obesity reached 20–30%. Undernutrition was higher among rural and indigenous children, whereas overweight predominated in urban and higher-income settings. Common dietary patterns included high consumption of ultra-processed foods and sugar-sweetened beverages, and insufficient intake of fruits, vegetables, and quality proteins. Maternal education, socioeconomic level, and school food environments were key determinants. **Conclusions:** Both malnutrition and overweight persist among Ecuadorian children, reflecting a nutrition transition influenced by socioeconomic and environmental factors. Context-specific public health actions are urgently needed, including school-based nutrition education, regulation of food marketing, improved access to affordable healthy foods, and community-level engagement to promote sustainable dietary habits. The included studies were mostly cross-sectional and often used non-validated dietary assessment tools, which may influence the reported prevalence estimates.

## 1. Introduction

Nutrition during childhood is essential for ensuring the overall physical, cognitive, and social development of individuals. During this stage, the body undergoes rapid changes that depend heavily on a proper and balanced diet. Adequate nutritional status is crucial for physical growth, academic performance, social skills, and the prevention of long-term chronic diseases such as obesity, diabetes, and cardiovascular disorders [1]. Eating habits acquired in early life have a profound impact on future health outcomes [2].

Globally, child malnutrition remains a pressing issue, manifesting as malnutrition, overweight, and obesity. According to the World Health Organization (WHO), approximately 144 million children under five experience stunting, 47 million suffer from wasting, and 38.3 million are overweight or obese. Furthermore, nearly 45% of child deaths are related to malnutrition [3]. These problems are exacerbated by global dietary transitions, driven by urbanization, shifts in food systems, and increased availability of ultra-processed foods [4].

In Latin America, while acute malnutrition has declined, overweight and obesity have risen sharply. In 2022, the prevalence of stunting in children under five was 11.5%, and obesity reached 8.6%, exceeding the global average of 5.6% [5]. This trend affects both urban and rural populations and highlights the coexistence of malnutrition and obesity in the region. Socioeconomic inequalities, limited access to healthy food, and aggressive marketing of processed products are major contributing factors [6].

Ecuador exemplifies this dual burden. Studies have documented growing dietary shifts among school-age children, marked by high consumption of sugary drinks, snacks, and processed foods, leading to increased overweight and obesity rates [7]. According to the National Health and Nutrition Survey (ENSANUT), 34% of Ecuadorian children are overweight or obese, while chronic malnutrition still affects 25.3% of children in rural areas [8]. These patterns highlight the need for targeted research to better understand eating habits and their relationship with nutritional outcomes across various regions of the country. Nutrition is considered a crucial factor in the physical, psychological, social, and cognitive development of children, particularly during early life stages [9]. Despite improvements in acute malnutrition, chronic malnutrition remains a major public health issue in Ecuador, affecting 27.2% and 23% of children under two and five and up, respectively, to 38% in rural settings [8].

In order to ensure the inclusion of regionally relevant evidence, the literature search was focused on SciELO, Dialnet, and ScienceDirect databases, which index the majority of studies conducted in Ecuador and other Latin American countries. These sources were complemented by manual searches of institutional repositories and reference lists to minimize coverage bias. Although this approach may limit international scope, it provides a more accurate representation of research produced within the Ecuadorian context, where much of the scientific output is not indexed in larger global databases.

Building upon this methodological framework, this review focused on children aged 1–11 years, encompassing both early childhood and pre-adolescence, periods during which dietary habits are established and strongly influence long-term health outcomes [10,11].

## 2. Materials and Methods

The methodological approach of this systematic review was guided by the PRISMA 2020 statement and supported by recommendations on systematic reviews in health research [12]. The review protocol was prospectively registered in PROSPERO (Registration Number: CRD420251080987).

### 2.1. Data Sources and Search Strategy

A comprehensive literature search was conducted in three databases—SciELO, Dialnet, and ScienceDirect—selected for their broad coverage of Spanish-language journals and Latin American research. Manual searches of the reference lists of included studies and relevant reviews were also performed to identify additional records not indexed in these databases.

The search covered studies published between 2018 and 2025 in English or Spanish, as studies published before 2018 were scarce and often used outdated nutritional assessment methods not aligned with current WHO growth standards and dietary evaluation tools. The databases were accessed in August 2025.

Search terms included combinations of “dietary habits,” “food consumption,” “feeding practices,” “nutritional status,” “malnutrition,” “undernutrition,” “overweight,” “obesity,” and “BMI,” in both English and Spanish. Boolean operators (AND, OR) were used to optimize retrieval of studies covering the full spectrum of nutritional conditions, from undernutrition to excess weight.

Search equations:

Spanish: (“hábitos alimenticios” OR “conductas alimentarias”) AND (“estado nutricional”) AND (“niños” OR “infancia”) AND (“Ecuador”).

English: (“dietary habits” OR “eating behaviors”) AND (“nutritional status”) AND (“children” OR “childhood”) AND (“Ecuador”).

The search focused on SciELO, Dialnet, and ScienceDirect because these databases provide comprehensive coverage of Latin American and Spanish-language journals, which are most likely to report Ecuador-specific data. Manual searches of reference lists complemented database searches to ensure inclusion of relevant studies.

### 2.2. Eligibility Criteria

The review focused on Ecuadorian children aged 1–11 years, as this period captures early childhood through preadolescence before the onset of puberty, which can influence dietary habits and nutritional status [10,11].

Inclusion criteria:Published between 2018 and 2025;Spanish or English language;Quantitative or mixed-methods studies;Ecuadorian school-age children;Explicit analysis of dietary habits and nutritional status.

Exclusion criteria:Studies on infants (<1 year), adolescents (>11 years), adults, or elderly;Conducted outside Ecuador;Review articles or purely theoretical studies.

Two independent reviewers (K.S.M. and C.I.-G.) screened titles, abstracts, and full texts. Disagreements were resolved by consensus or, when necessary, with a third reviewer (M.G.).

### 2.3. Data Extraction and Analysis

A standardized form was used for data extraction, including study characteristics (year, region, design, sample size, age range), dietary assessment methods, anthropometric indicators, and main nutritional outcomes. Extraction was conducted independently by two reviewers (K.S.M. and C.I.-G.), with discrepancies resolved by discussion and consensus or consultation with a third reviewer (M.G.).

From a total of 3542 records identified, 346 duplicates were removed. After screening titles and abstracts, 3021 studies were excluded for not meeting age, country, or design criteria. A total of 175 full-text records were reviewed in detail, and 17 studies met all eligibility criteria and were included in the final synthesis. The selection process is presented in the PRISMA flow diagram (Figure 1).

Due to heterogeneity in study design, age range, dietary assessment tools, and outcome measures, a quantitative meta-analysis was not feasible. Findings were synthesized narratively following Synthesis Without Meta-analysis (SWiM) guidelines, with descriptive summaries including ranges and medians where comparable.

### 2.4. Quality Assessment

Methodological quality was independently appraised by two reviewers using the CASPe checklist. Studies scoring ≥70% of the checklist criteria were classified as high quality, and those scoring 50–69% as moderate quality. Scores below 50% were considered low quality. Disagreements were resolved by consensus.

Most studies were rated as moderate to high quality, though limitations were noted regarding dietary assessment methods and representativeness of samples. Detailed quality assessments are presented in Appendix A.

## 3. Results

A total of 17 studies were included in this review; all were conducted in Ecuador. The studies analyzed dietary habits and nutritional status among school-age children aged 1–11. Methodologically, all studies employed descriptive cross-sectional designs with quantitative approaches, utilizing anthropometric evaluations and surveys or questionnaires to assess food intake.

A summary of the quality assessment is provided in Appendix A. Most studies met over 70% of CASPe criteria; however, limitations were observed in sampling representativeness and in the use of non-validated dietary questionnaires, which may affect the generalizability and comparability of findings across studies.

### 3.1. Overview of the Studies

Table 1 shows the main characteristics of the included studies: author, year, study objective, sample, methods, key findings, and nutritional status.

### 3.2. Patterns Observed

Across most studies, frequent consumption of sugary drinks, processed snacks, and refined flours was observed. Simultaneously, low consumption of fruits, vegetables, dairy, and whole grains was reported. Breakfast skipping, irregular meal patterns, and sedentary behaviors were also recurrent [15,16,17,18,20,21,22,23,24,25,26,27].

Rural children faced higher rates of malnutrition and stunting due to limited food access, low dietary diversity, and poverty. Conversely, urban schoolchildren showed higher overweight and obesity prevalence [22,24,26,28].

Overall, the included studies reveal a dual burden of malnutrition among Ecuadorian children. Stunting and malnutrition remain prevalent in rural areas, while overweight and obesity are more frequent in urban populations. Dietary patterns are characterized by high consumption of sugar-sweetened beverages, ultra-processed snacks, and refined carbohydrates, alongside low intake of fruits, vegetables, and high-quality proteins. Irregular meal patterns, breakfast skipping, and sedentary behaviors further exacerbate these nutritional challenges.

## 4. Discussion

The findings of this systematic review reveal a clear and consistent association between dietary habits and nutritional status among Ecuadorian children aged 1–11 years. Both forms of malnutrition—undernutrition and overweight/obesity—persist as major public health concerns, reflecting the double burden of malnutrition described across Latin America [6,29,30,31].

When stratifying the evidence by population characteristics, it becomes evident that undernutrition remains more prevalent among rural and indigenous children, while overweight and obesity are concentrated in urban and higher-income groups. These patterns mirror Ecuador’s broader socioeconomic and geographic inequalities. Such differences emphasize that aggregated national statistics can mask important disparities. Therefore, context-specific and regionally adapted interventions are needed to address these dual challenges effectively. These disparities are consistent with regional trends reported in Latin America, where social inequalities and gaps in public health policies continue to drive the coexistence of malnutrition and obesity [32].

Several included studies, such as Méndez et al. [13] and Guanoluisa et al. [20], documented high rates of both stunting and malnutrition coexisting with increasing overweight and obesity among school-aged children. These findings align with the National Survey on Child Malnutrition (ENDI 2023) [31], which reported alarming rates of poor anthropometric indicators in both urban and rural areas. Together, these results suggest that Ecuadorian children face overlapping nutritional risks related to inadequate diet quality, limited food diversity, and lifestyle changes associated with urbanization.

Poor dietary quality emerged as a dominant factor across studies. A consistent pattern of excessive intake of sugar-sweetened beverages, fried snacks, and ultra-processed foods was observed [15,16,17,18,25], combined with low consumption of fruits, vegetables, and micronutrient-rich foods such as iron, zinc, and vitamins A and D. This nutritional imbalance reflects the ongoing nutrition transition characteristic of middle-income countries, where traditional diets are being replaced by Westernized food patterns [4,33,34]. Moreover, low physical activity levels—with nearly 45% of children not meeting national recommendations [31]—further exacerbate the risk of overweight and metabolic imbalance. These findings are consistent with global data from the World Health Organization, which highlight obesity and overweight as growing challenges among children worldwide [35].

Socioeconomic determinants play a central role. Maternal education, household income, and food insecurity were repeatedly identified as significant predictors of nutritional outcomes [14,22,36]. Rural and indigenous communities continue to show higher prevalence of chronic malnutrition, as also evidenced by ENDI 2023 [31]. Environmental and contextual factors, such as limited access to clean water and poor sanitation, were frequently associated with undernutrition, particularly in low-income settings [37,38]. Conversely, exposure to energy-dense, low-nutrient foods and insufficient regulation of school food environments were linked to rising overweight rates. According to ENDI 2023, 67% of Ecuadorian children consume snacks at school at least three times per week, often without nutritional guidance [31].

Using the CASPe checklist, most studies were classified as moderate to high quality, meeting over 70% of the methodological criteria. However, an important pattern emerged: studies reporting higher malnutrition prevalence tended to have lower quality scores, indicating that some of these associations should be interpreted cautiously. In contrast, studies addressing overweight and obesity were generally of higher methodological quality, lending greater confidence to those findings. This integration between study quality and results strengthens the interpretive depth of the review and addresses one of the main concerns raised by reviewers.

Although a meta-analysis was not feasible due to methodological heterogeneity, a narrative synthesis identified clear subgroup trends:

Rural and indigenous children: higher rates of stunting and micronutrient deficiencies, linked to limited dietary diversity and persistent food insecurity.

Urban and higher-income children: greater prevalence of overweight and obesity, driven by sedentary behaviors and consumption of processed foods.

Socioeconomic vulnerability: strong correlation between low maternal education, poverty, and inadequate dietary practices.

These patterns illustrate the need for differentiated, equity-based strategies, as national-level averages underestimate disparities among vulnerable populations.

Beyond dietary and socioeconomic factors, recent research in Ecuador has highlighted the contributions of intestinal parasitic infections, poor sanitation, and limited access to safe water as structural determinants of child malnutrition [37,38]. Such multidimensional causes require integrated, multisectoral public health strategies.

This systematic review has several limitations. First, the literature search was restricted to three databases (SciELO, Dialnet, and ScienceDirect). This decision reflects the predominance of Ecuador-focused research in Latin American repositories and was complemented by manual searches of reference lists and institutional databases to minimize coverage bias. Although this may limit international reach, it ensures the inclusion of contextually relevant studies conducted in Ecuador. Most included studies were cross-sectional, and some used non-validated dietary assessment tools, which may affect the accuracy of reported prevalence rates. Additionally, rural and indigenous populations were underrepresented, limiting the generalizability of the findings.

Despite these limitations, the review provides one of the most comprehensive and up-to-date syntheses of evidence on dietary habits and nutritional status in Ecuadorian children, integrating methodological quality assessment and subgroup analysis.

## 5. Conclusions

This systematic review demonstrates a strong and consistent relationship between dietary habits and nutritional status among Ecuadorian children aged 1–11 years. The double burden of malnutrition—the coexistence of undernutrition and overweight—remains a critical challenge that reflects deep socioeconomic inequalities and the ongoing nutrition transition in Ecuador.

Public health implications: addressing this dual burden requires multisectoral and context-specific interventions, including:

Implementation of comprehensive nutrition education programs adapted to urban, rural, and indigenous settings.

Regulation of school food environments, particularly limiting the sale and marketing of ultra-processed, high-sugar foods.

Expansion of access to affordable, nutrient-rich foods through agricultural and social policies targeting low-income households.

Promotion of physical activity and family-centered health education to establish lifelong healthy habits.

Research priorities:

Future investigations should focus on:

Conducting nationally representative and longitudinal studies to monitor trends in child nutrition.

Validating dietary assessment tools suitable for Ecuadorian children.

Incorporating equity-based frameworks to better capture disparities among rural, indigenous, and urban populations.

By addressing these gaps, Ecuador can strengthen evidence-based policymaking to improve child nutrition and reduce inequalities. Ensuring adequate nutrition during early life is not only a matter of individual health but also a cornerstone for the country’s sustainable development and human capital growth.

## Figures and Tables

**Figure 1 nutrients-17-03608-f001:**
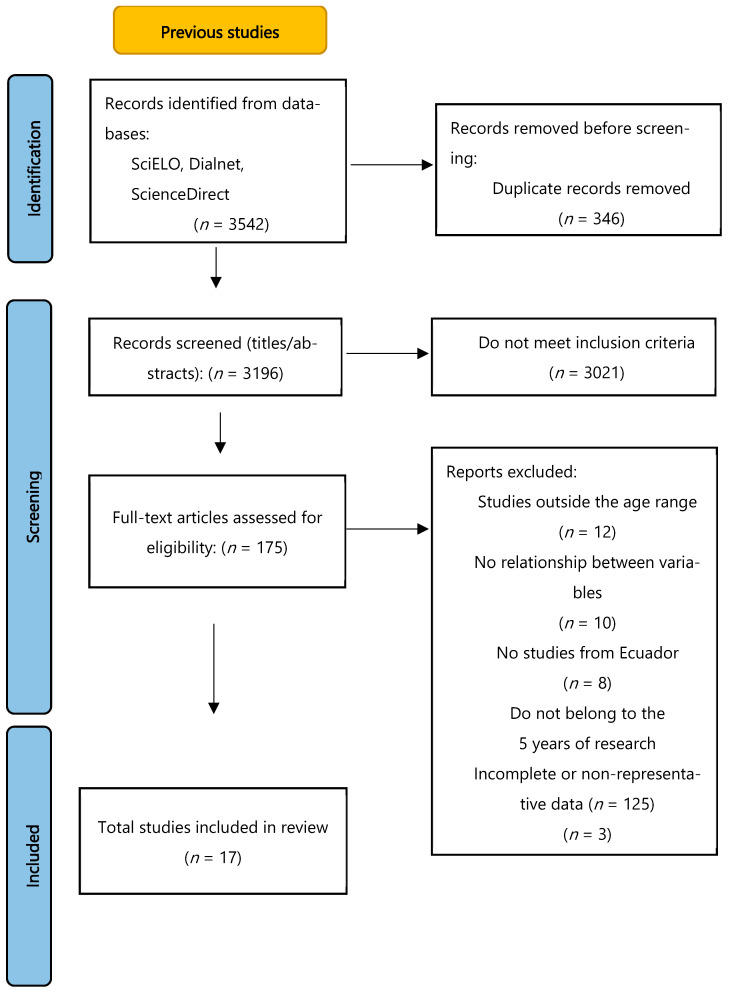
PRISMA 2020 flow diagram for systematic review of studies assessing dietary habits and nutritional status in Ecuadorian children.

**Table 1 nutrients-17-03608-t001:** Characteristics of the included studies (*n* = 17).

Author/Year	Type of Study	Population/Country	Intervention	Variables	Measures	Results
Ocaña & Sagñay. [9]	Cross-sectional	Children aged 0–3, Ecuador	Assessment of the relationship between nutritional status and cognitive development.	Malnutrition, growth, weight.	Statistical data analysis.	60% of children with moderate malnutrition, 23% with severe malnutrition. Malnutrition is the main factor interfering with cognitive development.
Méndez et al. [13]	Quantitative, cross-sectional	84 children (1–5 years), Ecuador	Dietary intake assessment using a 24 h recall	Macronutrients, height/age, weight/age.	45.24% short stature; 17.86% overweight/obesity; 74.36% protein malnutrition.	Dual burden of malnutrition: short stature combined with overweight and obesity.
Pacheco et al. [14]	Non-experimental, descriptive, and cross-sectional design.	90 children (6–11 years old) treated at the Arca Continental company in Quito, Ecuador.	Identification of the relationship between nutritional status and cardiometabolic risk.	Weight, height, and weight status.	66.67% of the children were of normal weight, 23.33% were overweight, and 8.89% were obese. Cardiometabolic risk was found in 36.67% of cases.	A moderate positive association was found between the children’s nutritional status and cardiometabolic risk.
Díaz & Da Costa [15]	Descriptive, cross-sectional, correlational study	125 parent-student pairs, Ecuador.	Characteristics of the eating habits and nutritional status of preschool children	Age, eating habits, food availability, weight, and height.	Nutritional status was poor due to low consumption of high biological value foods and high consumption of low nutritional quality foods.	Nutritional status was poor due to consumption of foods of high biological value (3.97%) and high consumption of foods of low nutritional quality (4.70%).
Caizaluisa, Quishpi & Pucha [16]	Descriptive and transversal	798 children (5–11 years old) Pichincha, Ecuador	Analysis of the relationship between eating habits and nutritional status.	BMI, consumption of processed foods and vegetables.	61.7% of the children were of normal weight, 19% were overweight, and 9.9% were obese.	Changing eating habits from childhood can prevent overweight and obesity, so a multidisciplinary strategy that includes health and education is necessary.
Peralta et al. [17]	Observational, analytical, and cross-sectional study	960 schoolchildren	Analysis of the relationship between academic performance and nutritional status	Social stratum, BMI, age.	8.7% were malnourished, 12.3% were obese, and 16.6% were overweight. A statistical association between academic performance and nutritional status.	The study found a significant correlation between academic performance and nutritional status of students Sayausí Millennium Educational Unit.
Vera, Zambrano & Ronquillo [18]	Quantitative and field-based	41 parents	Analysis of knowledge about eating habits	Eating habits.	A large proportion of parents were aware of appropriate eating habits, but most were unfamiliar with the school menu.	The study concluded that parents generally had some awareness of the problem, but children were not exempt from eating innutritious foods due to a lack of parental control.
Pozo & Vargas [19]	Cross-sectional and descriptive study	100 children from the SABIE Medical Center in Riobamba, Ecuador.	Analysis of the relationship between eating habits, nutritional status, and dyslipidemia.	Height/age, weight/age, weight and BMI/age.	Anthropometric data were significantly associated with diet-induced dyslipidemia.	Poor eating habits had a significant impact on the tendency toward overweight and obesity and on the presence of dyslipidemia.
Guanoluisa et al. [20]	Quantitative descriptive and retrospective	394 children and adolescents	Identification of nutritional status.	Weight, height, and body mass index.	High prevalence of overweight (12%) and obesity (23%).	A clear direct relationship between the consumption habits of households and the high rates of overweight and obesity, especially the consumption of saturated fats at an early age.
Sánchez et al. [21]	Cross section	252 schoolchildren aged 8 to 11	Assessment of nutritional status	Consumption, eating habits and practices.	Almost a quarter of the population surveyed is overweight and more than 28% is obese.	More than half of the 8- to 11-year-olds studied are overweight or obese, and families are not fostering healthy eating habits.
Hidalgo [22]	Quantitative with cross-section	215 children aged 0 to 3 from Salasaka, Ecuador	Analysis of the correlation between mothers’ educational level and children’s nutritional status	Weight, height, nutritional status.	15% of the children were underweight, 31% were stunted, 13% were malnourished, and 4% were overweight.	The study highlights the direct impact of a mother’s education on her child’s health and emphasizes the importance of a collaborative, multidisciplinary approach.
Álvarez et al. [23]	Quantitative, descriptive, cross-sectional cohort	476 children between 6 and 13 years old in Los Ríos, Ecuador	Identification of eating habits and nutritional status.	Breakfast consumption, consumption of PAE foods, number of meals per day.	37% of respondents skip breakfast, 52% of children eat breakfast from the PAE, and 43% eat breakfast 5 times a day.	The need for healthy eating guidelines that include mealtimes, portion sizes, marketing regulations, and monitoring of students’ nutritional status.
Escandón, Bravo & Castillo [24]	Descriptive, cross-sectional, and quantitative	104 children ages 3 to 5 from Azogues, Ecuador	Assessment of nutritional status	Age, weight, carbohydrate and processed food intake.	9.61% were underweight, 79.92% were normal weight, 11.53% were overweight, and 1.92% were obese.	Most of the children were well-nourished, but those with an insufficient body mass index had nutritional and social factors that hindered adequate development.
Gómez & Crespo [25]	Non-experimental quantitative cross-sectional and descriptive	36 clinical histories	Determination of nutritional status and its relationship with eating habits	Sociodemographic variables, nutritional status factors and anthropometric characteristics.	Nutritional status was related to their eating habits.	The determining factors of child malnutrition are the low level of education of the head of household and family income below the minimum wage.
León et al. [26]	Cross-sectional field research	Patients aged 5 to 9 years treated at Babahoyo General Hospital, Ecuador	Validation of a 2019 JCLA school instrument according to factors associated with malnutrition	Sociodemographic characteristics, nutritional assessment, family history of overweight and obesity.	Prevalence of low-nutrient diets in 55.6% of the samples.	A family’s current lifestyle affects their nutritional status. Consumption of low nutrient-dense foods directly impacts quality of life.
Darroman et al. [27]	Quantitative	345 children aged 1 to 10 from Los Ríos, Ecuador	Determination of the nutritional status of vulnerable groups.	The most common diseases, access to local health services, food reception, and processing.	High prevalence of malnutrition, with 72% of the children having nutritional and metabolic disorders.	Most people do not eat the recommended amounts of grains, dairy, vegetables, seafood, and fruits.
Pazmiño, Heredia & Yánez [28]	Descriptive—transversal	200 children from a rural parish in Chimborazo, Ecuador	Identification of risk factors and evidence of malnutrition	Height, age, BMI, weight, sex, factors associated with diet, health status, frequency of consumption of vegetables and fruits.	76% of the children had growth difficulties and 52% showed signs of malnutrition.	Children’s nutrition is determined primarily by family habits and the people who care for the child.

## Data Availability

Not applicable.

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
