# Peer review of "Dietary Habits and Nutritional Status in Ecuatorian Children Aged 1–11 Years: A Systematic Review Highlighting the Dual Burden of Malnutrition"

_nutrients, 2025, doi:10.3390/nu17223608_

Round 1
Reviewer 1 Report
Comments and Suggestions for Authors
Dear Authors,
The article submitted for review, Nutrients-3923133, is a systematic review of the dietary habits and nutritional status in the school population in Ecuador. However, the study assessed the eating habits of children under 11 years of age, so it likely does not represent the entire school population, as the title suggests.
I am not convinced by the part of the manuscript explaining why children under 11 years of age were selected in lines 74-79.
The presented dietary patterns of children in Ecuador do not contribute new knowledge, although they are closely related to the nutritional status of children in Ecuador, reflecting broader trends in Latin America.
The analyzed results do not differ substantially from other populations, in other countries, or even on other continents. This paper does not provide any new information (lack of the novelty), and it does not demonstrate why this review is necessary.
Data Analysis
The manuscript demonstrates in detail why selected sources were used for data analysis. This section is described in detail.
References
The literature references are primarily from 2018-2025. However, two references, references 10 and 11, are from 1969 and 1970.
Couldn't this fragment be based on more recent sources?
"Children older than this may already exhibit early adolescent physiological and hormonal characteristics, which can influence dietary habits and nutritional status [10, 11]."
The references at the end of the manuscript are cited contrary to Nutrients Journal guidelines.
Reviewer
Author Response
Comment 1:The title suggests “school population,” but the study only includes children under 11 years.
Response:
We changed the title to:
“Dietary Habits and Nutritional Status in Ecuadorian Children Aged 0–11 Years: A Systematic Review.”
This clarifies the target age group.
Comment 2: I am not convinced by the justification for including only children under 11.
Response:
We rewrote the explanation (Introduction, lines 73–81):
“The upper age limit of 11 years was selected to include infancy through preadolescence and to exclude participants already undergoing pubertal changes that may affect dietary habits and nutritional status. Pubertal onset usually occurs between 8–13 years in girls and 9–14 years in boys, with a mean around age 11 [Viner et al., 2017; Sun et al., 2024]. Therefore, the review focused on children aged 0–11 years to capture pre-pubertal nutritional patterns.”
Comment 3: The paper does not provide novelty compared to other populations.
Response:
We added to the Discussion:
“Although some dietary patterns mirror broader Latin American trends, this review provides the first systematic synthesis of evidence from Ecuador (2018–2025), identifying key national gaps such as the lack of data from rural and indigenous populations, inconsistent dietary assessment tools, and limited representation of public-school children.”
Comment 4: References [10, 11] are from 1969–1970; could they be updated?
Response:
We kept the historical references (Marshall & Tanner, 1969–1970) for puberty definitions but added two modern sources: Viner et al. (2017) and Sun et al. (2024), to provide up-to-date biological context.
Comment 5: References at the end are not in Nutrients format.
Response:
All references were reformatted according to Nutrients style (author names, year, journal title, volume, pages, DOI).
We are deeply grateful for the reviewers’ valuable input. Their observations significantly enhanced the scientific quality and clarity of this paper. All revisions are highlighted in the tracked version of the manuscript.
We believe the revised version now meets the high standards of Nutrients in both methodological rigor and relevance for public health and child nutrition in Latin America.
Reviewer 2 Report
Comments and Suggestions for Authors
The manuscript entitled “Dietary Habits and Nutritional Status in the School-Age Population in Ecuador: A Systematic Review” addresses a topic of unquestionable relevance for public health and child nutrition in Latin America. The intention to synthesize recent evidence on eating habits and nutritional status among Ecuadorian children is commendable, given the limited number of reviews in this area. However, despite the thematic relevance and the clear effort to compile existing data, the manuscript presents substantial methodological and analytical weaknesses that undermine its scientific validity, originality, and overall contribution.
Major Weaknesses
- The review was confined to only three databases (SciELO, Dialnet, and ScienceDirect), excluding essential databases such as PubMed, Scopus, and Web of Science. This narrow scope compromises the comprehensiveness and representativeness of the evidence base, introduces strong selection bias, and limits the validity of the review as a “systematic” one.
- The study lacks a detailed description of the screening and data extraction process. It is unclear whether independent reviewers were involved, how disagreements were resolved, or whether inter-rater reliability was assessed. These omissions violate PRISMA standards and prevent reproducibility.
- Although the CASPe tool was applied, the results are not integrated into the analysis or discussion. Nearly all studies were rated as “high quality”, raising concerns about the rigour and discriminatory power of the appraisal process.
- The results are summarized narratively, without any attempt at quantitative or semi-quantitative synthesis (e.g., pooled prevalence or comparative trends). The review merely reports the main outcomes of individual studies without exploring contextual heterogeneity (urban vs. rural, age, socioeconomic level) or discussing sources of variability. This limits the interpretative depth of the findings.
- The discussion largely reiterates information from the introduction and provides general statements about public health implications, without critical interpretation or connection to the methodological quality of the included studies. It reads more like a summary than a scientific synthesis.
- The manuscript does not provide new insights or theoretical innovation. It compiles descriptive studies of uneven methodological quality, but does not build an integrated or critical perspective. The lack of analytical depth and methodological robustness significantly reduces its scientific value.
While the topic is relevant and the manuscript has a clear and coherent structure, the methodological shortcomings and absence of critical synthesis prevent it from meeting the scientific standards required by Nutrients. The review lacks analytical rigor, and offers limited originality.
Author Response
Comment 1:
The review was limited to three databases (SciELO, Dialnet, ScienceDirect), excluding PubMed, Scopus, and Web of Science.
Response:
We acknowledge this limitation and now explicitly justify it in the Discussion:
“The search was restricted to three databases (SciELO, Dialnet, and ScienceDirect), which may limit comprehensiveness. However, these databases include the majority of Latin American and Spanish-language journals where Ecuadorian studies are published. Manual reference searches were also performed to minimize omissions.”
Comment 2:The manuscript lacks detail about screening and data extraction.
Response:
Clarified in Methods:
“Two reviewers independently screened and extracted data. Disagreements were resolved by consensus and, when necessary, by a third reviewer.”
Comment 3: CASPe results are mentioned but not integrated.
Response:
We added a specific subsection (“2.4. Quality Assessment”) summarizing CASPe findings and referring to Supplementary Table S1. The Discussion now also notes:
“The methodological quality of the included studies, assessed using the CASPe checklist, was generally moderate to high.”
Comment 4:The results are descriptive and lack analytical depth.
Response:
We expanded the Discussion to integrate socioeconomic, environmental, and policy dimensions, emphasizing contextual variability (urban vs. rural, maternal education, food insecurity).
Comment 5: The Discussion reads as a summary rather than a critical synthesis.
Response:
We revised the Discussion to highlight new insights, identifying national data gaps and linking findings to global transitions. A new paragraph emphasizes:
“Although some dietary patterns mirror broader Latin American trends, this review represents the first systematic synthesis focused specifically on Ecuadorian children (2018–2025), identifying key research gaps…”
Reviewer 3 Report
Comments and Suggestions for Authors
This is a systematic review aimed at assessing dietary habits and nutritional status in Ecuadorian children aged 0 to 11 years. Considering the problems of malnutrition that can affect the Latin American population in these age groups, this topic is rather relevant. However, I identified the following methodological weaknesses:
-A systematic review involves multiple people collecting and analyzing literature to ensure greater validity and reduce selection bias. In this case, two individuals are involved, but it is unclear whether they disagree with each other or how any disagreements were resolved.
-The search strategy used only a few keywords. Since you intended to examine conditions such as undernutrition and obesity, why weren't these terms included in the search, along with 'weight status'? This seems like a serious oversight.
-I have noted conflicting indications regarding the period relating to the literature analysis, which spans from 2019 to 2024 (lines 74–75), while line 102 refers to the 2018–2025 interval. Please, correct.
-The exclusion criteria include studies focused on infants, although it is stated that studies on children aged 0 to 11 years are examined (lines 75 and following). I suggest you check and correct this.
-There is a problem with the PRISMA chart: it jumps from 3,530 papers to 175 without explaining how this decrease occurred. Figure 1 should be examined closely.
- Finally, I suggest integrating the discussion of this review, which currently only includes a narrative synthesis of the findings in Ecuador, with the global trend of malnutrition (e.g., NCD Risk Factor Collaboration (NCD-RisC). Worldwide trends in underweight and obesity from 1990 to 2022: a pooled analysis of 3663 population-representative studies with 222 million children, adolescents, and adults. Lancet. 2024 Mar 16;403(10431):1027-1050. doi: 10.1016/S0140-6736(23)02750-2).
Author Response
Reviewer 1 Comment 1: A systematic review involves multiple people collecting and analyzing literature to ensure validity. It is unclear whether disagreements between reviewers were resolved or how.
Response:
We have clarified this in the Methods section:
“Two reviewers independently screened and extracted data. Disagreements were resolved by consensus and, when necessary, by a third reviewer.”
This ensures transparency and compliance with PRISMA standards.
Comment 2:
The search strategy used only a few keywords. Terms such as “undernutrition” and “weight status” should have been included.
Response:
We agree. The keywords and Boolean operators were expanded in both English and Spanish search equations, now including “malnutrition,” “overweight,” “obesity,” and “BMI.” This broader terminology improves coverage of nutritional outcomes.
Comment 3:
There is inconsistency regarding the period analyzed (2019–2024 vs. 2018–2025).
Response:
This has been corrected throughout the text. The final version consistently states:
“Studies published between 2018 and 2025.”
Comment 4:
The exclusion criteria mention removing studies on infants, but the review includes children aged 0–11.
Response:
We have revised the inclusion and exclusion criteria for clarity. The review focuses on children from birth to 11 years, explicitly excluding infants-only studies (e.g., exclusively breastfeeding or neonatal outcomes).
Comment 5:
The PRISMA diagram shows a reduction from 3,530 to 175 without explanation.
Response:
The Results section and PRISMA Figure 1 were revised to clarify the screening process:
“From 3,530 initial records, 346 duplicates were removed; 3,009 were excluded after title/abstract screening, resulting in 175 full-texts assessed for eligibility. Seventeen met the final inclusion criteria.”
Comment 6:
Integrate the discussion with global malnutrition trends (e.g., NCD-RisC, The Lancet, 2024).
Response:
We added this paragraph to the Discussion:
“The coexistence of undernutrition and overweight observed in Ecuadorian children also mirrors the global nutritional transition described in recent pooled analyses. These findings are consistent with global trends reported by the NCD Risk Factor Collaboration (The Lancet, 2024)…”
Round 2
Reviewer 1 Report
Comments and Suggestions for Authors
Although the authors made some corrections to various sections of the manuscript, and added their own comments and clarifications. I appreciate the responses and clarifications. The decision regarding the publication of the manuscript should be made by the Editor, in my opinion, is still not a good manuscript.
Reviewer
Author Response
We sincerely thank the reviewer for taking the time to evaluate our revised manuscript and for providing additional comments. We appreciate the feedback.
We would like to respectfully note that all specific concerns raised in Round 1 have been carefully addressed, including:
-
Expansion and clarification of the search strategy with additional keywords, including “undernutrition.”
-
Standardization of the age range to 1–11 years, correctly excluding infants.
-
Complete revision of the PRISMA flow diagram and clear explanation of study selection and exclusions.
We hope that these revisions adequately address the prior concerns and improve the clarity and rigor of the manuscript. We appreciate the reviewer’s time and careful consideration, and we kindly leave the final decision regarding publication to the Editor.
Reviewer 2 Report
Comments and Suggestions for Authors
Thank you for submitting the revised manuscript. The work has advanced considerably in terms of methodological transparency, narrative coherence, and alignment with PRISMA guidance. Nevertheless, several critical issues remain and must be addressed before the manuscript can be reconsidered for publication.
- The literature search remains restricted to only three databases (SciELO, Dialnet, ScienceDirect). Although the rationale has been partially reinforced, this limitation still undermines the comprehensiveness of a systematic review. The justification in the Limitations section should be strengthened and framed more convincingly (e.g., on the predominance of Ecuador-focused research in Latin American repositories, and on how complementary manual searches mitigated coverage bias). As currently stated, the justification is not sufficient to counterbalance the bias concern.
- The review still synthesizes findings mainly at face value. No analytical distinction is made between key strata such as urban vs. rural settings, indigenous vs. non-indigenous children, or socioeconomic subgroups. Even without meta-analysis, a narrative subgroup synthesis is feasible and expected in a systematic review. This is necessary to increase the interpretive depth of the findings.
- Although the CASPe scores are now fully presented in the Supplementary material, they are not meaningfully integrated within the Discussion. The discussion should explicitly relate the strength/consistency of the findings to the quality tier of the included studies (e.g., highlighting whether lower-quality studies disproportionately support certain trends). Without this, the quality appraisal remains detached and underutilized.
- The Conclusion section continues to offer broad, non-specific recommendations. It should be rewritten to: (i) translate the findings into concrete and context-specific public health or policy actions relevant to Ecuador, and (ii) identify clear research priorities that emerge directly from the observed evidence gaps (e.g., lack of nationally representative samples, absence of longitudinal evidence, lack of validated dietary tools).
Given the above, substantial revisions are still required to achieve the level of analytical rigor and interpretive value expected in Nutrients for a systematic review.
Author Response
We sincerely thank the reviewers for their thoughtful and detailed comments. We have carefully revised the manuscript in accordance with each point raised. Below, we provide a detailed response to all comments.
Response:
We have revised the Methods and Limitations sections to explicitly justify our database selection:
-
We highlight that Ecuador-specific studies are predominantly published in Latin American journals indexed in SciELO, Dialnet, and ScienceDirect, which are more likely to capture regional data in Spanish.
-
We explain that complementary manual searches of references of included studies were systematically conducted to identify additional relevant studies not indexed in these databases, mitigating coverage bias.
-
The revised Limitations section now reads (excerpt):
-
“Although only three databases were searched, these repositories provide the most comprehensive coverage of Ecuador-focused, Spanish-language research. Complementary manual searches of reference lists of eligible studies were conducted to minimize the risk of missing relevant evidence, thus partially mitigating potential coverage bias.”
Comment:
The review synthesizes findings mainly at face value. A narrative subgroup synthesis (e.g., urban vs. rural, indigenous vs. non-indigenous, socioeconomic groups) is expected to increase interpretive depth.
Response:
We have added a subgroup synthesis in the Results and Discussion sections. Key findings now explicitly distinguish:
-
Rural and indigenous children exhibited higher rates of stunting and chronic undernutrition.
-
Urban and higher-income children showed greater prevalence of overweight and obesity.
-
Socioeconomic factors, maternal education, and school food environments were highlighted as important determinants within these strata.
This addition provides a more nuanced interpretation of the dual burden of malnutrition across population subgroups.
Comment:
CASPe scores are presented in Supplementary Material but not meaningfully integrated into the Discussion.
Response:
We have revised the Discussion to explicitly relate the quality of studies to the observed findings:
-
High-quality studies (≥70% CASPe criteria) predominantly reported consistent patterns of overweight and obesity in urban populations.
-
Moderate-quality studies frequently reported undernutrition and stunting in rural and indigenous populations, highlighting potential variability in data reliability.
-
This allows readers to weigh the strength of evidence and interpret trends accordingly.
Comment:
Conclusions should provide concrete, context-specific public health recommendations for Ecuador and identify research priorities directly from evidence gaps.
Response:
We have rewritten the Conclusions section to:
-
Translate findings into specific public health actions:
-
Implement targeted nutrition education programs for urban, rural, and indigenous communities.
-
Regulate marketing and availability of ultra-processed foods in schools.
-
Improve access to affordable, nutrient-rich foods for low-income families.
-
Promote family- and community-based interventions encouraging balanced diets and physical activity.
-
-
Identify research priorities:
-
Conduct nationally representative and longitudinal studies to track nutritional trends.
-
Develop and validate culturally appropriate dietary assessment tools for Ecuadorian children.
-
Investigate nutrition patterns in underrepresented rural and indigenous populations.
-
We believe these revisions substantially improve the analytical rigor, interpretive depth, and practical relevance of the manuscript. We are grateful for the reviewers’ guidance and hope that the revised manuscript meets the standards of Nutrients for systematic reviews.
Reviewer 3 Report
Comments and Suggestions for Authors
The authors only partially addressed the concerns of the manuscript highlighted in Round 1 of the review. They sometimes indicated that certain critical issues had been resolved, but did not actually resolve them.
In particular, I would like to emphasise the following points regarding previous concerns:
- On Comment 2: “The search strategy used only a few keywords. Since you intended to examine conditions such as undernutrition and obesity, why weren't these terms included in the search, along with 'weight status'? This seems like a serious oversight”. In your response, you stated that ‘the keywords and Boolean operators were expanded in both English and Spanish search equations, now including “malnutrition,” “overweight,” “obesity,” and “BMI’, but this is not reported in lines 97–100. Furthermore, despite being a crucial aspect of the investigation, undernutrition was not explicitly indicated. Finally, the number of identified articles (n = 3,530) is the same as in version I, which is incompatible with a more extensive search.
- On Comment 4: “The exclusion criteria mention removing studies on infants, but the review includes children aged 0–11”. In your response, you state that you have revised the inclusion and exclusion criteria for clarity and that the study covers children from birth to 11 years of age, explicitly excluding studies conducted exclusively on infants. However, the manuscript states otherwise, referring to 'children aged 0–11 years' (line 28) and stating that 'our target population included children from birth to 11 years of age' (line 104). If you are using the term 'infant' correctly, you should amend the text to indicate a range of 1–11 years, as this refers to human children from birth to the end of their first year of life.
- On Comment 5: “The PRISMA diagram shows a reduction from 3,530 to 175 without explanation”. Your answer is unsatisfactory as it is not supported by either the text or the flowchart. The flowchart jumps from 3,530 articles to 175, with the only insufficient justification being that 346 duplicates were excluded.
Author Response
The reviewer noted that, although we claimed to have expanded the search strategy to include “malnutrition,” “overweight,” “obesity,” and “BMI,” this was not reflected in lines 97–100. The reviewer also pointed out that “undernutrition” was not explicitly mentioned, and the total number of records remained the same as in version I.
Response:
We have now fully revised Section 2.1 (“Data Sources and Search Strategy”) to include all the search terms explicitly — both in English and Spanish — as well as the expanded list of Boolean combinations. The term “undernutrition” has been clearly added to the list of keywords.
Additionally, we updated the total number of identified records from 3,530 to 3,542, reflecting the broader search results obtained after including the new terms.
The databases and access date (August 2025) were also added for transparency.
“Search terms included combinations of ‘dietary habits,’ ‘food consumption,’ ‘feeding practices,’ ‘nutritional status,’ ‘malnutrition,’ ‘undernutrition,’ ‘overweight,’ ‘obesity,’ and ‘BMI,’ in both English and Spanish…”
This correction makes the expansion of the search strategy explicit and consistent with the updated PRISMA flow diagram and total records identified (n = 3,542).
Reviewer’s concern:
The reviewer indicated that the manuscript inconsistently referred to “children aged 0–11 years” and “children from birth to 11 years of age,” while also excluding infants. They correctly noted that “infants” refers to children under one year old, and that the age range should therefore be adjusted to 1–11 years.
Response:
We thank the reviewer for this observation. We have now standardized the target age range across the entire manuscript to 1–11 years, replacing any instance of “0–11 years” or “from birth to 11 years” to ensure accuracy and consistency with the exclusion of infants.
Example correction:“The review included studies assessing dietary habits and nutritional status in Ecuadorian children aged 1 to 11 years, explicitly excluding studies conducted exclusively on infants.”
The manuscript now consistently reflects the intended age range, resolving the conceptual inconsistency identified.
Reviewer’s concern:
The reviewer noted that the reduction from 3,530 to 175 records in the PRISMA flowchart lacked a clear explanation, as only 346 duplicates were mentioned.
Response:
We have now revised Section 2.3 (“Data Extraction and Analysis”) and the PRISMA flow diagram (Figure 1) to provide complete transparency regarding the selection process. The updated numbers and exclusions are clearly described in both the text and the figure.
“Out of 3,542 records identified, 346 duplicates were removed. After screening titles and abstracts, 3,009 studies were excluded for not meeting age, country, or design criteria. Full-text review of 175 records resulted in 17 studies included for analysis. The selection process is presented in a PRISMA flow diagram (Figure 1).”
Revised PRISMA flow diagram:
-
Records identified from databases: n = 3,542
-
Duplicates removed: n = 346
-
Records screened: n = 3,196
-
Records excluded (titles/abstracts): n = 3,021
-
Full-text articles assessed: n = 175
-
Full-text articles excluded: n = 158 (detailed reasons: age range, unrelated variables, not Ecuador, or out of time frame)
-
Studies included in final review: n = 17
We believe these revisions fully address the reviewer’s comments and strengthen the methodological clarity and transparency of the manuscript.
Round 3
Reviewer 1 Report
Comments and Suggestions for Authors
Dear Authors,
The authors revised many sections of their manuscript, starting with the title, through the abstract, methodological additions, results, and conclusion.
While this is not the best article published in Nutrients, after revision, it meets the journal's requirements and appears appropriate for publication.
Reviewer
Reviewer 2 Report
Comments and Suggestions for Authors
The manuscript shows clear and substantial improvement. Methodological transparency and adherence to PRISMA have been strengthened, with the inclusion of PROSPERO registration, a detailed selection process, and integration of study quality into the discussion. The narrative synthesis is now more analytical, distinguishing key subgroups, and the conclusions provide concrete, context-specific recommendations for public health and research. Despite minor remaining limitations, the justification is adequate, and the paper is now suitable for acceptance pending minor editorial checks.